# Beyond Labels and Topics: Discovering Causal Relationships in Neural Topic Modeling

## ABSTRACT

Topic models that can take advantage of labels are broadly used in identifying interpretable topics from textual data. However, existing topic models tend to merely view labels as names of topic clusters or as categories of texts, thereby neglecting the potential causal relationships between supervised information and latent topics, as well as within these elements themselves. In this paper, we focus on uncovering possible causal relationships both between and within the supervised information and latent topics to better understand the mechanisms behind the emergence of the topics and the labels. To this end, we propose Causal Relationship-Aware Neural Topic Model (CRNTM), a novel neural topic model that can automatically uncover interpretable causal relationships between and within supervised information and latent topics, while concurrently discovering high-quality topics. In CRNTM, both supervised information and latent topics are treated as nodes, with the causal relationships represented as directed edges in a Directed Acyclic Graph (DAG). A Structural Causal Model (SCM) is employed to model the DAG. Experiments are conducted on three public corpora with different types of labels. Experimental results show that the discovered causal relationships are both reliable and interpretable, and the learned topics are of high quality comparing with seven start-of-the-art topic model baselines.

## CCS CONCEPTS

• **Information systems** → **Document topic models**; **Data mining**; **Web mining**.

## KEYWORDS

Causal Relationships Discovery, Neural Topic Model, Structural Causal Model

**ACM Reference Format:**
Anonymous Author(s). 2024. Beyond Labels and Topics: Discovering Causal Relationships in Neural Topic Modeling. In *Proceedings of Make sure to enter the correct conference title from your rights confirmation emai (WWW '24)*. ACM, New York, NY, USA, 9 pages. https://doi.org/XXXXXXX.XXXXXXX

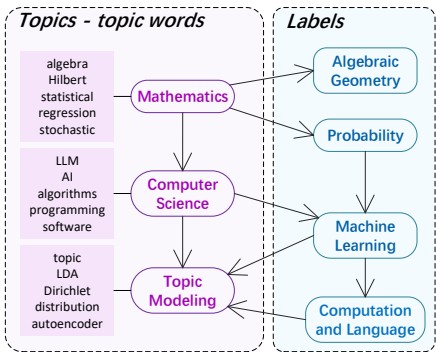

**Figure 1: Examples of the causal relationships (directed edges) between the manually annotated labels and the latent topics (nodes), as well as within these elements themselves.**

## 1 INTRODUCTION

Topic modeling is a family of text mining techniques aimed at automatically discovering representative and semantically interpretable topics from textual data [6, 24]. Topic models are wildly

used in a variety of AI tasks, including web mining and information retrieval [14, 32]. In recent years, the incorporation of supervised information has gained attraction in topic modeling. The supervised information mainly contains semantic labels and text categories. For the first branch, topic models typically map each semantic label to a specific topic or a cluster of topics. The semantics of these labels then serve to guide the interpretability and relevance of the discovered topics [18, 27, 42]. Parallel to this, another significant branch of topic models views the supervised information as categories or outcomes of texts. This perspective has been instrumental in advancing several natural language processing (NLP) tasks, such as document classification [8, 29].

However, despite the ability of these models to exploit supervised information to enhance the discovery of semantically related topics, most existing topic models ignore the complex relationships between supervised information and latent topics, as well as within these elements themselves. To alleviate the problem, in this paper, we propose a novel neural topic model that can jointly extract the latent topics and discover potential causal relationships between supervised information and the latent topics. Causal relationships in textural data and its supervised information are widespread in the real-world. The reason why we choose "causal relationship" to model the complex relationships mentioned above is that compared to the common relationships in topic modeling, such as correlation [5, 34, 38] and hierarchical relationships [9, 26, 42], the applicable scenarios of causal relationships are wider and the relationships that can be described are more comprehensive.

Figure. 1 shows a toy instance of the causal relationships between the supervised information and the latent topics, as well as within these elements themselves. The left side of Figure. 1 displays the discovered latent topics with top-5 topic words in topic modeling, while the right side denotes some manually annotated labels. Causal relationships are represented as the directed edges. Taking

the node *"Topic Modeling"* as an example, the topic *"Topic Modeling"* is influenced not only by the topic *"Computer Science"*, but also by the label *"Machine Learning"* and *"Computation and Language"*. Recognizing the causal relationships is crucial for understanding the mechanisms behind the emergence of certain topics within the context of specific labels. However, the causal relationships between the supervised information and the latent topics are rarely modeled by existing topic models. Furthermore, the causal relationships cannot be replaced by correlation or hierarchical relationships without missing semantic information.

In order to automatically discover the causal relationships discussed above, in this paper, we propose a novel neural topic model, called Causal Relationship-Aware Neural Topic Model (CRNTM). CRNTM takes the textual data and the supervised information as inputs and jointly learns the causal relationship between and within the supervised information and the latent topics. Specifically, CRNTM is built upon the variational autoencoder (VAE) framework with a Dirichlet distribution prior [7]. It encodes the input texts and the supervised signals into low-dimensional latent topical representations and then learns the inherent causal relationships both between and within the supervised information and latent topics. To jointly learn the three types of causal relationships, we consider both the latent topics of documents and the supervised information as nodes in a Directed Acyclic Graph (DAG), and the causal relationships can be represented as the directed edges. We adopt a Structural Causal Model (SCM) [17, 39, 41, 43] to learn the causal relationship DAG. SCM is a type of strategy to model causal relationships between variables in causal inference based on the theory of structural equation modeling (SEM). By integrating SCM as a causal relationship learning module, the model can discover the inherent causal relationships both between and within the supervised information and latent topics, and generate the causality enhanced representations of the variables. We further introduce some regularization functions to optimize the causal relationship matrix and make it conform to the properties of causal relationships, including the directed acyclicity of the DAG, the information transmission of parent-child nodes in causal relationships, and the counterfactual regularization.

We conduct experiments on three public corpora from the real-world with different types of supervised information, such as age ratings, document categories and annotated tags. We compare the discovered topics with seven start-of-the-art topic model baselines in terms of topic coherence, topic uniqueness and topic quality. The experimental results show that the topics learned by our proposed CRNTM are of high quality. Furthermore, we visualize the discovered causal relationships between the variables and show their reliability and interpretability.

Our main contributions can be summarized as follows:

- We propose a novel neural topic model that can capture the potential relationships between supervised information (i.e. labels) and latent topics, as well as within these elements themselves, simultaneously.
- We employ the Structural Causal Model to jointly model the causal relationships between supervised information and the latent topics, since causality is widespread and covers most situations in practice.

## 2 RELATED WORK

In this section, we briefly review state-of-the-art neural topic models, along with a discussion on the interplay between supervised information and topics in topic modeling.

### 2.1 Neural Topic Models

Topic modeling is wildly used in automatically uncovering representative topics from corpora [6, 14, 15, 31]. In recently years, neural topic modeling [19, 30] has attracted much attention thanks to the development of deep learning.

ProdLDA is the first autoencoding variational Bayes (AEVB) [13] based topic model, which uses a Laplace approximation to represent the Dirichlet distribution prior. Gaussian Softmax (GSM) [19] uses the Gaussian Softmax distribution to parameterize the latent multinomial topic proportion of each document. W-LDA [21] introduces Wasserstein autoencoders (WAE) [33] to topic modeling, allowing topic proportions to follow the Dirichlet prior. Sparse Dirichlet variational autoencoder (DVAE Sparse) [7] implements the rejection sampling variational inference (RSVI) as the reparameterization function of the Dirichlet distribution prior in VAE based neural topic modeling. TAN-NTM [23] uses an LSTM to extract contextual information and an attention mechanism to identify words relevant to each topic in topic modeling. Coordinated Topic Modeling (CTM) [1] uses a set of well-defined topics as prior knowledge for easily understandable representation. NSEM-GMHTM [9] enhances hierarchical topic modeling by incorporating a Gaussian mixture prior for improved sparse data handling, and explicitly representing both hierarchical and symmetrical topic relationships through dependency matrices and nonlinear structural equations.

Most recently, the incorporation of pre-trained language models into topic modeling has provided contextualized semantic embedding. Examples include embedded topic model (ETM) [10], CombinedTM [3], enhanced guided LDA model [35], BERT-Flow-VAE [16] and BERTopic [12].

### 2.2 Supervised Information and Topic Models

The integration of supervised information into topic modeling, which blends structured, labeled data with the unsupervised extraction of latent topics, is an increasingly pivotal area of study and has the potential to greatly enhance our understanding and utilization of text corpora. In topic modeling, supervised information are usually utilized to guide the semantic structure and enrich the quality of the topics leading to improved model performance [36, 40]. Existing topic models that can leverage supervised information can be broadly divided into two types: 1) Supervised information is used as a guidance of the semantic of the topics (or a subset of topics), representative models include Labeled LDA (LLDA) [27], Partially LDA (PLDA) [28], Topic Attention Model [37], JoSH [18] and supervised BERTopic [12]. 2) Supervised information, typically in the form of category labels, is correlated with the topic proportion vectors of each document [8, 12]. Models such as SCHOLAR [8] offer flexible incorporation of metadata into neural topic modeling. HIMECat [42] is a neural topic model that can integrate the label hierarchy, metadata, and text signals for document categorization under weak supervision. HSTM [29] is designed to model the structure in text

concurrently capturing heterogeneity in the relationship between text and outcomes of documents.

However, none of these models take into consideration the causal relationships between supervised information and latent topics. To the best of our knowledge, our work is the first one that can uncover potential causal relationships between the supervised information and latent topics, and jointly model the causal relationships within these elements themselves.

# 3 CAUSAL RELATIONSHIP-AWARE NEURAL TOPIC MODEL

This section provides an in-depth introduction of the proposed CRNTM and its learning strategy. The objective of CRNTM is to jointly discover interpretable causal relationships between and within the supervised information and latent topical representations from the corpus and manual labels. CRNTM is built upon the variational autoencoder (VAE) framework [7, 19]. Initially, the model encodes the input texts and the supervised information into low-dimensional latent topical representations based on the Dirichlet distribution prior. Subsequently, the model endeavors to identify the causal relationships between and within the learned latent topical representations and the supervised signals to enrich the topical embeddings with causality via a Structural Causal Model (SCM). An illustration of the model architecture is provided in Figure. 2, showcasing the process from latent topical representation learning to causal relationship identification.

## 3.1 Latent Topical Representation Learning

To capture the latent topical representation of a document alongside its supervised information, we use a two-phase representation learning method. First, a pre-encoding phase learns topical representations that encapsulate the essential semantic information within the input texts. Then, we jointly encode the latent variables of the documents and the supervised information. This approach provides a comprehensive understanding of both the latent topics and the supervised information inherent in the document.

*3.1.1 Pre-encoding for the documents.* Assume the input corpus contains a set of $D$ documents, denoted as $\{x_d\}_{d=1}^D$, where $x_d \in \mathbb{R}^V$ represented as a vector in a $V$-dimensional space using a bag-of-words model (BoW), with $V$ being the size of the vocabulary. Each document $x_d$ is associated with a set of supervised information, denoted as $l_d$.

Different from most VAE based neural topic models [2, 19, 44], where the variational distribution are drawn from the Gaussian distribution, in this paper, we use the Dirichlet distribution as a prior for the latent topical representations. The input document $x_d$ is transformed into the Dirichlet parameter $\alpha$, which can be represented as:

$$\lambda = MLP(x_d),$$
$$\alpha = \log(1 + e^\lambda),$$ (1)

where $MLP(\cdot)$ denotes a multilayer perceptron layer, which can transfer the input $x_d \in \mathbb{R}^V$ to latent topical variables, $\lambda \in \mathbb{R}^K$. $K$ denotes the number of the latent topics. $\alpha$ is the Softplus function of $\lambda$ for smoothness.

Then, we can draw a topic proportion $\theta_d$ from a Dirichlet distribution parameterized by $\alpha$. Since the Dirichlet distribution does not support non-central differentiable reparameterization, we adopt the proposal function of a rejection sampler as the reparameterization function based on the rejection sampling variational inference (RSVI) [7, 20]. In RSVI, a complex or unknown probability distribution (referred to as the target distribution), denoted as $q(z; \alpha)$, can be sampled from an easier-to-sample proposal distribution, denoted as $r(z; \alpha)$, with a constant accept rate $M_\alpha$:

$$q(z; \alpha) \le M_\alpha r(z; \alpha).$$ (2)

As discussed in [4], a Dirichlet distribution with parameter vector $\alpha$ can be sampled from independent Gamma distributions with the same parameter $\alpha$. Therefore, the latent topical representation $z$ drawn from the Dirichlet distribution with parameter $\alpha$, i.e. $z \sim Dirichlet(\alpha)$, can be simulated from the distribution with Gamma-distributed random variables:

$$\tilde{z}_{d,k} \sim \Gamma(\tilde{\alpha}_k, 1), k = 1, ..., K,$$ (3)

Then latent topical representation $z$ can be computed through the simulated latent topical representation $\tilde{z}$:

$$z_{1:K} = \frac{1}{\sum_k \tilde{z}_{d,k}} (\tilde{z}_{d,1}, ..., \tilde{z}_{d,K})^\top \sim Dirichlet(\alpha_{1:K}).$$ (4)

For Gamma distribution, there exists an efficient rejection sampler [20]:

$$z = h_\Gamma(\epsilon, \alpha) := (\alpha - \frac{1}{3})(1 + \frac{\epsilon}{\sqrt{9\alpha - 3}})^3, \epsilon \sim s(\epsilon) := \mathcal{N}(0, 1),$$ (5)

where $\epsilon$ is the accepted sample in the rejection sampler.

Since the rejection sampler has higher acceptance rates for higher values of the parameter $\alpha$ in the Gamma distribution, we use a shape augmentation trick following the idea in RSVI [20] to solve the problem. Suppose $B$ is a positive integer. Then, $z$ can be expressed as: $z = \tilde{z} \prod_{i=1}^B u_i^{\frac{1}{\alpha+i-1}}$, $\tilde{z} \sim \Gamma(\alpha + B, 1)$ and the uniform random variable $u_i \overset{i.i.d.}{\sim} U[0, 1]$. Therefore, the above rejection sampling Eq. (3) can be redefined as $\tilde{z} \sim \Gamma(\alpha + B, 1)$, and the shape augmented Eq. (5) can be redefined as:

$$z = h_\Gamma(\epsilon, \alpha, B) := (\alpha + B - \frac{1}{3})(1 + \frac{\epsilon}{\sqrt{9(\alpha + B) - 3}})^3.$$ (6)

*3.1.2 Joint encoding for both texts and supervised information.* To leverage the supervised information inherent in the textual data and further explore causal relationships, we propose a joint encoding of the supervised information and latent variables of the documents.

In our model, the supervised information can take various forms such as labels or categories associated with the texts. The model imposes no restrictions on the type of supervised signals, allowing for great flexibility. These signals could include both quantifiable (numeric) and non-quantifiable (non-numeric) information. For example, quantifiable information can be converted into real numbers, while non-quantifiable information could be represented as binary variables. This versatility allows a wide range of data types to be incorporated into our model.

Specifically, we concatenate the latent topical variables of document $d$, denoted as $\alpha$, with the associated supervised information

**Figure 2: Network structure of CRNTM. The pre-encoding phase ingests textual data $x$ and learns to represent the essential semantic information as topical representations. The joint encoding phase combines the latent topical variables with the supervised information and encodes them into the same semantic embedding space. The causal relationship learning module uncovers potential causal relationships both between and within the latent topics and the supervised signals and enriches the representations with causality via a DAG. See more details in section 3.**

$l_d$, and encode them in a manner similar to the pre-encoding phase section 3.1.1:

$$\lambda^* = MLP(Contact(\alpha; l_d))), \qquad (7)$$

$$\alpha^* = \log(1 + e^{\lambda^*}), \qquad (8)$$

$$z^* = h_\Gamma(\epsilon^*, \alpha^*, B) := (\alpha^* + B - \frac{1}{3})(1 + \frac{\epsilon}{\sqrt{9(\alpha^* + B) - 3}})^3, \qquad (9)$$

where $l_d \in \mathbb{R}^S$ denotes the supervised signals of document $d$, and $S$ is the number of supervised signals. The comprehensive representation $z^*$ contains both the latent topic information from the text and the supervised information.

## 3.2 Causal Relationship Learning

We utilize a Structural Causal Model (SCM) [39] to learn the causal relationships between and within the latent representations and the supervised signals. The causal relationships between the variables can be modeled via the weighted adjacency matrix of the Directed Acyclic Graph (DAG), denoted as $A$. In CRNTM, both the latent topics and the supervised variables are considered as nodes in the causal relationship DAG. We arrange the topics and supervised variables as the first few nodes and the last few nodes in the DAG respectively. Then, the causal relationships between the latent topics, and between the supervised variables, can be represented by the upper left and lower right parts of the DAG's weighted adjacency matrix, respectively. Meanwhile, the causal relationships between topics and supervised variables can be represented by the remaining parts of the adjacency matrix. In this way, we can use the weighted adjacency matrix to clearly represent the above three types of causal relationships in supervised information and latent topics. The dimension of the weighted adjacency matrix of the DAG equals the total number of latent topics ($K$) and the supervised signals ($S$), i.e. $A \in \mathbb{R}^{(K+S)\times(K+S)}$. According to structural causal learning [41], we have the following linear structural equation model (SEM):

$$C_d^* = A^\mathsf{T} C_d^* + z_d^\blacklozenge = (I - A^\mathsf{T})^{-1} z_d^\blacklozenge, \qquad (10)$$

where $C_d^* \in \mathbb{R}^{(K+S)\times H}$ is the causal representation of document $d$, which denotes the causal relationships enhanced representations. $H$ is the dimension of the causal representations. $z_d^\blacklozenge = t(z_d^*)$, where $t(\cdot)$ is a linear transformation layer, and $z_d^\blacklozenge \in \mathbb{R}^{(K+S)\times H}$ is an extension of the latent topical representation, $z_d^*$, to make it contain more semantic information.

In order to enhance the directionality in causal relationships, the following constraints should be concerned: the causal representation of a node i) is not allowed to contain information from its non-parent nodes; and ii) ought to incorporate the representation information of its parent nodes to ensure the information transformation from parents to children. Therefore, we adopt the Mask Layer [22, 39] in CRNTM to implement the above two constraints:

$$C_{d,i} = g_i(A_i \circ C_d^*) + z_{d,i}^\blacklozenge, \qquad (11)$$

where $A_i$ denotes the $i^{th}$ column in the weighted causal adjacency matrix $A$, $C_{d,i}$ is the masked latent causal representation of the $i^{th}$ node (topic or supervised signal) in document $d$, and $\circ$ is the element-wise multiplication. The function $g_i(\cdot)$ is a mild nonlinear function for input variables to do self reconstruction.

To better understand Eq. (11), we can consider the following extreme cases.

- If $A_{j,i} > 0$ in matrix $A$, the $j^{th}$ node is a parent of the $i^{th}$ node. Then, $[A_i \circ C_d]_j \not\equiv \mathbf{0}$, and Eq. (11) can be expressed as a function of $C_{d,j}$: $C_{d,i} = G_i(C_{d,j}^*)$.
- If $A_{j,i} = 0$, $[A_i \circ C_d^*]_j \equiv \mathbf{0} \Rightarrow \forall G_i(\cdot), C_{d,i} \neq G_i(C_{d,j}^*)$.

The topic proportion $\theta_d$ can be computed through the causal topical representation $C_d$:

$$\theta_d = \text{softmax}(f(C_d)), \qquad (12)$$

where $f(\cdot)$ is a linear transformation layer, and $\theta_d \in \mathbb{R}^K$.

In the decoding step, we reconstruct the original input documents with the topic proportion $\theta_d$ and the topic distribution

$\beta \in \mathbb{R}^{K \times V}$. The reconstruction of a word $x_{d,n}$ in the text $x_d$ can be modeled as $x_{d,n} \sim Mult(\text{softmax}(\theta_d \cdot \beta))$.

## 3.3 Learning and Inference

Our proposed CRNTM takes the documents $\{x_d\}_{d=1}^D$ and the associated supervised signals $\{l_d\}_{d=1}^D$ as inputs to discover the causal relationships between and within the learned latent topical representations and the supervised signals. The generative model of document $d$ in CRNTM can be written as:

$$\mathbb{E}_{q(D)}\big[\sum_{d=1}^D \log p(x_d^1, x_d^2 | u_d)\big]$$
$$=\mathbb{E}_{q(D)}\big[\sum_{d=1}^D \log p(x_d^1)\big] + \mathbb{E}_{q(D)}\big[\sum_{d=1}^D \log p(x_d^2 | u_d)\big] \quad (13)$$
$$\geq \mathcal{L}_{D_{pre}} + \mathcal{L}_{D_{joint}},$$

where $x_d^1$ and $x_d^2$ are two independent copy of $x_d$, which are used in the reconstruction in the the pre-encoding phase and the joint encoding phase, respectively. $\mathcal{L}_{D_{pre}}$ denotes the evidence lower bound (ELBO) of the pre-encoding phase, and $\mathcal{L}_{D_{joint}}$ is the ELBO of the joint encoding phase.

$$\mathcal{L}_{D_{pre}} = \mathbb{E}_{q(D)}\big[\mathbb{E}_{q(z|x_d)}\big[\sum_{n=1}^N \log p(x_{d,n}|z_d)\big] \\ - D_{KL}(q(z_d|x_d)\|p(z_d))\big], \quad (14)$$

where $D_{KL}(\cdot\|\cdot)$ denotes the KL divergence.

According to RSVI, the distribution of the accepted sample $\epsilon$, $\pi(\epsilon; \phi)$, can be obtained by marginalizing over a uniform variable $u$ of the rejection sampler:

$$\pi(\epsilon; \alpha, B) = \int \pi(\epsilon, u; \alpha, B) du = s(\epsilon)\frac{q(h_\Gamma(\epsilon; \alpha, B))}{r(h_\Gamma(\epsilon; \alpha, B))}, \quad (15)$$

where $r(\cdot)$ is the proposal function for the rejection sampler.

Therefore, Eq. (14) can be written as:

$$\mathcal{L}_{D_{pre}} = \mathbb{E}_{q(D)}\big[\mathbb{E}_{\pi(\epsilon;\alpha,B)}\big[\sum_{n=1}^N \log p(x_{d,n}|h_\Gamma(\epsilon; \alpha, B))\big] \\ + \mathbb{E}_{\pi(\epsilon;\alpha,B)}\big[\log \frac{p(h_\Gamma(\epsilon; \alpha, B))}{q(h_\Gamma(\epsilon; \alpha, B)|x_d)}\big]\big], \quad (16)$$

For the joint encoding phase:

$$\mathcal{L}_{D_{joint}} = \mathbb{E}_{q(D)}\big[\mathbb{E}_{q(C_d|z_d,l_d)}\big[\sum_{n=1}^N \log p(x_{d,n}|C_d)\big] \\ - D_{KL}(q(C_d|z_d, l_d)\|p(C_d|l_d))\big]. \quad (17)$$

Moreover, to optimize the mask parameters in the Mask Layer, we need to minimize the following equation according to Eq. (11) [22]:

$$\mathcal{L}_m = \mathbb{E}(\sum_{i=1}^{K+S} \|C_i - g_i(A_i \circ C)\|^2). \quad (18)$$

Furthermore, the discovered causal adjacency matrix $A$ is supposed to be a DAG. On the basis of DAG-GNN [41], the causal adjacency matrix $A$ should satisfy the following condition:

For any $\rho > 0$, the graph is acyclic if and only if:

$$H(A) \equiv tr\big((I + \rho A \circ A)^{(K+S)}\big) - (K + S) = 0. \quad (19)$$

Since we assume that the latent topical representation and the supervised signals are causally related in this work, we further introduce a counterfactual regularization item following [17] to ensure the convincingness of the learned causal DAG structure. It is a commonsense that in a causal relationship, changing the cause leads to a change in the effect, but changing the effect does not lead to a change in the cause. Therefore, for the nodes in the causal DAG, the following equations holds true:

$$\text{causal direction} \quad : l_i \to C_j \Rightarrow C(l_i) \neq C(do(l_i)),$$
$$\text{anti-causal direction}: l_n \leftarrow C_m \Rightarrow C(l_n) = C(do(l_n)), \quad (20)$$

where $\to$ and $\leftarrow$ represent the direction of the causal relationship, and $do(\cdot)$ denotes the do-operation where we set $l_i \neq do(l_i)$.

Specifically, we further train a binary classifier $\mathbf{D}$ to distinguish the causal counterfactuals and the anti-causal counterfactuals. The counterfactual representation contrastive regularizer can be written as:

$$\mathcal{L}_{do} = \mathbb{E}[\mathbb{E}_{\Omega^+}(\mathbf{D}(C(do(l_i)))) + \mathbb{E}_{\Omega^-}(1 - \mathbf{D}(C(do(l_n))))], \quad (21)$$

where $\Omega^+ = \{l_i | l_i \in Parents(C_j), C_j \in \mathbf{C}, l_i \in \mathbf{l}\}$, $\Omega^- = \{l_n | l_n \in Children(C_m), C_m \in \mathbf{C}, l_n \in \mathbf{l}\}$.

To sum up, considering the above Eq. (16), Eq. (17), Eq. (19), Eq. (18), and Eq. (21), we get the overall loss function of the proposed model:

$$\mathcal{L} = -\mathcal{L}_{D_{pre}} - \mathcal{L}_{D_{joint}} + H(A) + \mathcal{L}_m + \mathcal{L}_{do}. \quad (22)$$

## 4 EXPERIMENTS

### 4.1 Experimental Setup

*4.1.1 Corpora.* We conduct experiments on three public corpora, including *Russian books*[1][11], *ArXiv*[2] and *StackSample*[3]. For each corpus, we tokenize the lowercased documents, remove the nltk stop words, and then perform the stemming step based on the nltk SnowballStemmer tool[4], respectively. The statistics of corpora are listed in Table 1.

**Table 1: The statistics of the corpora.**

| Corpora | Label type | Label | Train | Test | Voc |
|---------|-----------|-------|-------|------|-----|
| *Russian books* | age rating, genre | 37 | 4,492 | 1,000 | 10,000 |
| *ArXiv* | discipline category | 20 | 2,228,866 | 10,000 | 10,000 |
| *StackSample* | question tag | 20 | 821,724 | 10,000 | 10,000 |

*4.1.2 Baselines and Experimental Settings.* We compare the proposed model with seven state-of-the-art neural topic models, including GSM[5] [19], SCHOLAR[6] [8], DVAE[7] [7], HIMECat[8] [42],

[1]https://www.kaggle.com/datasets/oldaandozerskaya/fiction-corpus-for-agebased-text-classification
[2]https://www.kaggle.com/datasets/Cornell-University/arxiv
[3]https://www.kaggle.com/datasets/stackoverflow/stacksample
[4]https://www.nltk.org
[5]We use a PyTorch version modified from the author provided Tensorflow version: https://github.com/ysmiao/nvdm.
[6]https://github.com/dallascard/scholar
[7]https://github.com/sophieburkhardt/dirichlet-vae-topic-models
[8]https://github.com/yuzhimanhua/HIMECat

**Table 2: A comparison of the topic coherence (TC), topic unique (TU) and topic quality (TQ). We compute the mean value of each metric over top-5 and top-10 topical words, and higher value represents better performance. The best results are in bold. See more details in section 4.2.**

| Model | Russian books | | | | | | ArXiv | | | | | | StackSample | | | | | |
|---|---|---|---|---|---|---|---|---|---|---|---|---|---|---|---|---|---|---|
| | K = 20 | | | K = 50 | | | K = 20 | | | K = 50 | | | K = 20 | | | K = 50 | | |
| | TC | TU | TQ | TC | TU | TQ | TC | TU | TQ | TC | TU | TQ | TC | TU | TQ | TC | TU | TQ |
| GSM | 0.200 | 0.325 | 0.065 | 0.172 | 0.271 | 0.047 | 0.143 | 0.630 | 0.090 | 0.154 | 0.563 | 0.087 | 0.190 | 0.400 | 0.076 | 0.162 | 0.396 | 0.064 |
| SCHOLAR | 0.262 | 0.875 | 0.229 | 0.215 | 0.652 | 0.140 | 0.105 | 0.950 | 0.100 | 0.134 | **0.933** | 0.125 | 0.185 | 0.972 | 0.180 | 0.212 | **0.923** | 0.196 |
| DVAE | 0.333 | 0.940 | 0.313 | 0.316 | 0.712 | 0.225 | 0.301 | **0.998** | 0.300 | **0.355** | 0.897 | **0.318** | 0.462 | 0.958 | 0.443 | 0.438 | 0.853 | 0.374 |
| HIMECat | 0.160 | 0.951 | 0.152 | - | - | - | 0.206 | 0.980 | 0.202 | - | - | - | 0.311 | **0.990** | 0.308 | - | - | - |
| BERTopic | 0.134 | 0.752 | 0.101 | - | - | - | 0.249 | 0.782 | 0.195 | - | - | - | 0.274 | 0.793 | 0.217 | - | - | - |
| HSTM | 0.054 | **0.995** | 0.054 | 0.044 | **0.981** | 0.043 | 0.015 | 0.748 | 0.011 | 0.018 | 0.540 | 0.010 | 0.034 | 0.75 | 0.026 | 0.026 | 0.666 | 0.017 |
| NSEM-GMHTM | 0.196 | 0.838 | 0.164 | 0.198 | 0.768 | 0.152 | 0.026 | 0.765 | 0.020 | 0.038 | 0.738 | 0.028 | 0.110 | 0.695 | 0.076 | 0.112 | 0.687 | 0.077 |
| CRNTM | **0.351** | 0.920 | **0.323** | **0.328** | 0.709 | **0.233** | **0.361** | **0.998** | **0.360** | 0.339 | 0.910 | 0.308 | **0.503** | 0.978 | **0.492** | **0.501** | 0.846 | **0.424** |

**Table 3: Examples of the top-10 words per topic and the corresponding topic coherence (TC) values on *ArXiv*.**

| solar activity | astrochemistry | galactic composition | hardware | telecommunication | astronomy | particle physics | carcinology | Wi-Fi | NLP |
|---|---|---|---|---|---|---|---|---|---|
| CMEs | desorption | GCS | GPUs | multiple-input | Jupiter | seesaw | tumor | backhaul | multilingual |
| rope | Deuterium | Gyr | FPGAs | precoding | close-in | B-L | lung | downlink | cross-lingual |
| CME | gas-phase | metal-rich | FPGA | MIMO | planet | leptogenesis | breast | uplink | monolingual |
| reconnection | photodissociation | Fe/H | CPUs | multiple-output | TESS | vector-like | cancer | NOMA | low-resource |
| corona | CH3OH | bulge | HPC | beamform | super-earth | NMSSM | liver | D2D | bilingual |
| footpoint | HCN | gas-rich | CUDA | CSIT | extrasolar | CP-Even | malignant | QoS | NMT |
| magnetogram | isotopologues | galactocentric | Intel | downlink | GJ | MSSM | nodule | offload | BERT |
| eruption | r-process | mass-to-light ratio | Nvidia | OFDM | Neptune | slepton | prostate | relay | corpora |
| Alfvén | prestellar | Alpha/Fe | GPU | CSI | semi-major | sneutrino | lesion | 5G | BLEU |
| Hinode | H2 | globular | NISQ | multi-antenna | semimajor | R-Parity | histology | caching | PLMs |
| 0.50 | 0.34 | 0.40 | 0.41 | 0.54 | 0.42 | 0.37 | 0.48 | 0.39 | 0.46 |

supervised BERTopic[9] [12], HSTM[10] [29] and NSEM-GMHTM[11] [9].

We conduct experiments using a variable topic number of 20 and 50 for the baseline models and our proposed model across each of the three corpora. Following DVAE [7], we set the hidden units of our proposed model to 100, and a dropout rate of 0.25 is implemented. The Dirichlet prior is set to 0.01 and the shape augmentation parameter $B$ is set to 10 both as per the DVAE source code. We set the dimension of the causal topical representation $H$ to $\{1, 2, 4, 8, 16, 32, 64, 128\}$, and choose $H = 2$ for *Russian books*, $H = 32$ for *ArXiv* and $H = 128$ for *StackSample* according to the quality of the learned topics on each training set. We initialize the learning rate at 0.001 and set the batch size to 256. The Adam optimizer is employed to train our model, and the model's performance is monitored on a validation set, employing an early stopping strategy if no improvement is observed over 30 epochs. The parameter settings of the baseline models are kept consistent with those detailed in their respective original papers.

## 4.2 Evaluation on Topic Quality

In topic modeling, the quality of the learned topics (TQ) [21] is typically assessed from two perspectives: topic coherence (TC) [25]

and topic uniqueness (TU) [21]. Topic coherence can measure the semantic similarity between the top words within the same topic. In this paper, we use the normalized pointwise mutual information (NPMI) based topic coherence[12]. The topic coherence score for topic $k$ with top $N$ words can be computed by:

$$TC(k) = \sum_{j=2}^{N} \sum_{i=1}^{j-1} \frac{\log \frac{P(w_j, w_i)}{P(w_i)P(w_j)}}{-\log P(w_i, w_j)}. \quad (23)$$

On the other hand, topic uniqueness reflects the discriminative power of a topic in relation to others, indicating the extent to which a topic captures unique aspects of the corpus that are not covered by other topics. TU can be computed by:

$$TU = \frac{1}{K \cdot N} \sum_{k=1}^{K} \sum_{n=1}^{N} \frac{1}{cnt(n, k)}, \quad (24)$$

where $cnt(n, k)$ is the total number of times the $n^{th}$ word in topic $k$ appears in the top words across all the topics. The quantitative measure of topic quality is calculated as the product of these two factors: $TQ = TC \times TU$. This approach ensures that high-quality topics are both semantically meaningful and distinct from each other.

In the experiments, we compute the mean value of each metric over top-5 and top-10 topical words in the discovered topics. A

---

[9]https://github.com/MaartenGr/BERTopic
[10]https://github.com/dsridhar91/hstm
[11]https://github.com/nbnbhwyy/NSEM-GMHTM

[12]https://github.com/jhlau/topic_interpretability

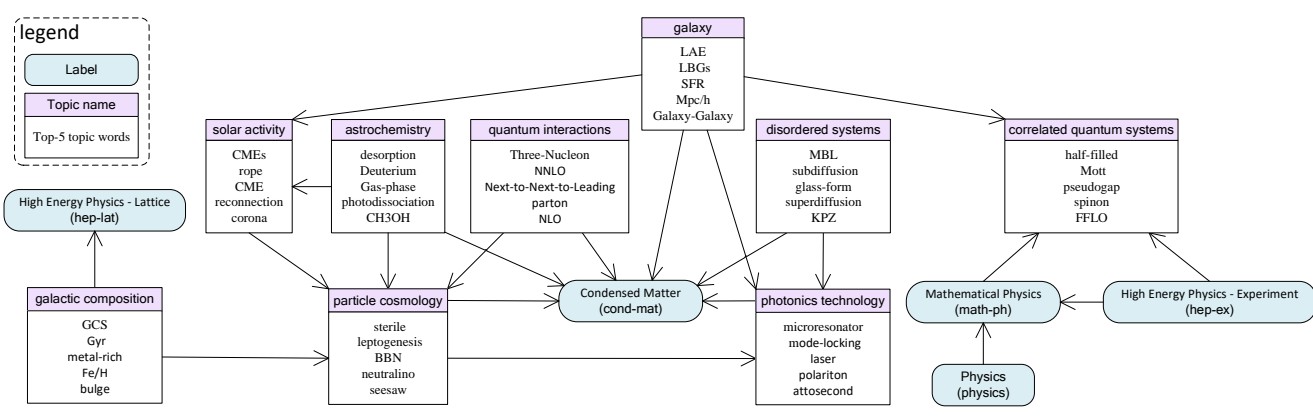

**Figure 3: Examples of the causal relationships between and within the supervised information and the latent topics. The topic names are manually assigned based on the top-5 words. The spatial relative positions between different variables are merely for illustrative purposes, and bear no relation to the causal relationships between the variables. In other words, these relationships do not form a hierarchical structure. See more details in section 4.3.**

higher value indicates better results for all the above three metrics. The experimental results are shown in Table 2. The missing value for HIMECat and BERTopic is attributable to the fact that the number of topics in these models is inherently linked to the total number of supervision signals. Therefore, we adjust the topic count to match the label count of the corresponding corpora.

The results demonstrate that our proposed CRNTM outperforms other models in most cases, particularly concerning two crucial metrics - topic coherence and topic quality. Despite a slightly lower topic uniqueness compared to SCHOLAR, HIMECat and HSTM in some cases, the topic coherence and overall metric topic quality of CRNTM significantly exceed those of these three models. Actually, topic uniqueness typically becomes a valuable reference primarily when the topics generated by the model are semantically meaningful. Hence, TU is more meaningful when the topics are both unique and coherent, underlining the importance of balancing these two metrics in topic modeling.

CRNTM outperforms all other baselines on topic quality metric, with the exception of DVAE in some cases. Compared to DVAE, our model achieves better results in terms of topic coherence, topic uniqueness, and topic quality, except some occasional cases where it falls slightly short. This indicates that our model can effectively learn high-quality topics. Additionally, our model is capable of discovering causal relationships between supervised information and latent topics, an achievement that other baselines fail to accomplish. This unique capability further enhances the robustness and interpretability of our model, providing an essential tool for deeper understanding in topic modeling.

We further display the top-10 words of some example topics learned from *ArXiv*. The topic names in the first line are manually assigned based on the topical words. For ease of presentation and comprehension, the top words have been lemmatized from the stem forms in the second line. The real numbers below the words represent topic coherence values. These examples demonstrate the semantic coherence and interpretability of the topics discovered by our proposed model.

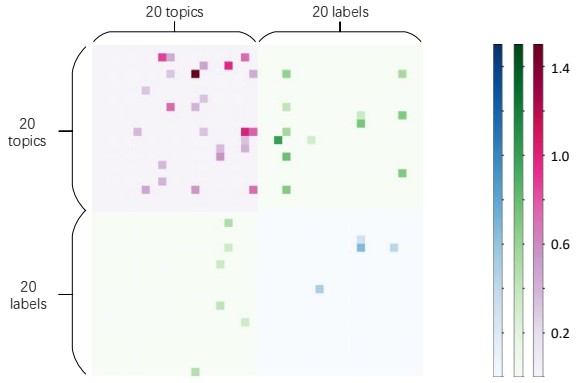

**Figure 4: The weighted adjacent matrix of the causal relationship DAG on *ArXiv* with 20 topics. See more details in section 4.3.**

## 4.3 Causal Relationships between Supervision Information and Latent Topics

We demonstrate the discovered causal relationships between the supervised information and the latent topics to show the ability of our model in causal relationship discovery. We extract the causal relationships from the learned weighted adjacency matrix by using a thresholding value 0.3 to rule out cycle-inducing edges following NOTEARS [43], since a small threshold suffices to rule out cycle-inducing edges. Figure. 4 is the learned weighted adjacent matrix of the causal relationship DAG on *ArXiv* with 20 topics. The first 20 nodes are the learned latent topics, and the last 20 ones are the supervised variables. In the adjacent matrix, the causal relationships between different topics, between supervised information, and between supervised information and the topics are denoted with different colors. Figure. 4 confirms that our model is capable of simultaneously constructing causal relationship between supervised signals and topics, as well as within each of them individually.

To further demonstrate the learned causal relationships, we select several discipline category (supervised information) and topics from the DAG on *ArXiv* with 50 topics shown in Figure. 3. From the examples, we can see that the discovered causal relationships are credible and interpretable to a certain extent. For example, for the causal relationship between the supervised information and the latent topics, the causal chain from the topic *"quantum interactions"* to the label *"cond-mat"* reflects the impact of quantum effects on the macroscopic state of matter. The topic *"photonics technology"* → the label *"cond-mat"* is reasonable because new photonics technology can be used to study and control the macroscopic state of matter. For the causal relationships between topics, the causal chain from the topic *"solar activity"* to the topic *"particle cosmology"* reveals that it may be reasonable for solar activity to affect particle behavior and distribution in the universe, because solar activity produces a large number of high-energy particles and radiation, which can affect particle cosmology. The topic *"galactic composition"* → the topic *"particle cosmology"* shows the effect of the composition of the galaxy on particle cosmology, for the composition of the galaxy can affect the behavior and distribution of particles within it. Furthermore, the proposed model can also uncover causal relationships between supervised signals, such as causal relationships among label *"hep-ex"*, *"math-ph"* and *"physics"*.

## 4.4 Ablation Study

To study the contribution of each component of our model, we consider the following three types of components:

- The DAG: The restricted condition on the directed acyclic nature of DAGs ($H(A)$); the counterfactual regularization in causal relationships ($\mathcal{L}_{do}$); and the Mask Layer on causal structure ($\mathcal{L}_m$).
- The prior distribution: the prior distribution of the document vectors, Gaussian distribution or Dirichlet distribution.
- The encoding phase: whether to use the pre-encoding phase.

We ablate different components in nine cases: without the Mask Layer loss (#2 in Table 4); without the condition on the directed acyclicity (#3); without the counterfactual regularization regularization (#4); without directed acyclicity and counterfact (#5); without the above three parts (#6); without the DAG network structure and its related loss, and the model degrade into the baseline DVAE (#7); without the pre-encoding phase (#8); replace the Dirichlet distribution with Gaussian distribution (#9); replace the Dirichlet distribution with Gaussian distribution and without the DAG network structure, and the model degrade into the baseline GSM (#10).

Table 4 shows the topic quality results of the ablation study experiment on *StackSample* under 50 topics. The complete CRNTM model achieves the best performance across most metrics, and achieves the best overall topic quality. The removal of each part of optimizing the DAG leads to a noticeable drop in the performance. This indicates that our assumption of a directed acyclic causal relationship existing between the supervised information and the latent topics is reasonable. Incorporating the directed acyclic causal relationship into topic modeling can effectively enhance the topic discovering capability of the model and guide the model to better

**Table 4: A comparison results of the ablation experiments on *StackSample* under 50 topics. See more details in section 4.4.**

| # | Model | $K = 20$ | | | $K = 50$ | | |
|---|-------|------|------|------|------|------|------|
| | | TC | TU | TQ | TC | TU | TQ |
| 1 | CRNTM | **.503** | **.978** | **.492** | **.501** | .846 | **.424** |
| 2 | w/o $\mathcal{L}_m$ | .495 | .972 | .481 | .463 | .872 | .404 |
| 3 | w/o $H(A)$ | .426 | .912 | .389 | .444 | .872 | .387 |
| 4 | w/o $\mathcal{L}_{do}$ | .490 | .953 | .467 | .404 | .831 | .336 |
| 5 | w/o ($H(A) + \mathcal{L}_{do}$) | .465 | .925 | .430 | .458 | .894 | .409 |
| 6 | w/o ($\mathcal{L}_m + H(A) + \mathcal{L}_{do}$) | .441 | .940 | .415 | .454 | .884 | .401 |
| 7 | w/o DAG (DVAE) | .462 | .958 | .443 | .438 | .853 | .374 |
| 8 | w/o pre-encoding phase | .362 | .962 | .348 | .340 | **.957** | .325 |
| 9 | $\mathcal{N}(\cdot)$+DAG | .307 | **.978** | .300 | .246 | .942 | .232 |
| 10 | GSM | .190 | .400 | .076 | .162 | .396 | .064 |

understand and capture the underlying structure of the corpus, leading to more accurate and robust topic modeling. Moreover, the prior distribution of the document vectors is confirmed to be a important role in discovering interpretable topics, for models under the Dirichlet distribution outperform than that under the Gaussian distribution.

Furthermore, the model without the pre-encoding phase demonstrates a significant reduction in topic coherence compared to the complete model. This indicates the effectiveness of our model's pre-encoding phase, which is capable of mapping the crucial semantic information from the input documents to the latent topic space to provide ample semantic information in discovering causal relationships. The pre-encoding phase ensures that the VAE framework can strike a balance between learning the semantic information of the latent topics and the causal relationship structure of the topics. This allows the model to capture the intricate relationships between topics and their semantic, leading to a more coherent and interpretable topic model. In summary, each component of CRNTM contributes significantly to its performance.

## 5 CONCLUSION

In this paper, we undertake an exploration into the causal relationships between and within the supervised information and latent topics in neural topic modeling. We propose Causal Relationship-Aware Neural Topic Model (CRNTM), a novel approach designed to automatically unravel significant causal relationships in supervised information and latent topics, while concurrently discovering high-quality topics, thereby enhancing the overall interpretability of the model. We conceptualize these causal relationships as directed edges within a Directed Acyclic Graph (DAG), treating both supervised information and latent topics as nodes. We employ a Structural Causal Model (SCM) to imbue the representations of the supervised information and the latent topics with causality, modeling these interactions within the causal relationship DAG. The experimental results confirm the reliability and interpretability of the causal relationships uncovered. Moreover, they underscore the high quality of the learned topics.

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

Received 20 February 2007; revised 12 March 2009; accepted 5 June 2009

