# OpenReview forum: "Beyond Labels and Topics: Discovering Causal Relationships in Neural Topic Modeling"
_ACM.org/TheWebConf/2024/Conference — TheWebConf24_

### Official Review · Reviewer_WaQX · 2023-11-15

**Novelty:** 5
**Technical Quality:** 5

**Review:**

This work proposes a new neural topic model that is capable of identifying causal relationships between/within supervised information and discovered topics. The three components (pre-encoding, joint encoding, and causal relationship learning) are employed to learn latent topics and corresponding casual relationships. Overall, the proposed idea is interesting and the proposed method is well justified. The clear presentation helps with readability. There is room for improvement in discussing motivations for the idea and methods. It is not sure if the codes will be provided for reproducibility.

**Questions:**

1. It would be a general question, but personally, it is not clear to me what "influence" means in the identified causal relationships. For instance, there is an example saying "A topic Computer Science influences a topic Topic Modeling". Simply speaking, does this imply that Topic Modeling can be a topic because of Computer Science? Some discussion on the meaning of causal relationships between topics would be helpful.

2. The authors argue that identifying causal relationships is more useful in many scenarios than merely getting correlation and hierarchical relationships. Could the authors give more specific examples of how the topic modeling with causal relationships is practically useful?

3. Along the same line, the role and necessity of supervised information in this work is not clear. How does the supervised information affect the performance of the proposed model?

4. (Minor) Can the authors provide justification or motivation for using Dirichlet distributions instead of Gaussian distributions to model latent topics?

5. (Comment) Sections 3.2 and 3.3 were a bit overwhelming with a lot of complex concepts and theoretical background. I believe it would be nice to supplement the intuitive explanations. and save the specific formulas and such for the appendix.

6. (Comment) Authors are strongly encouraged to provide the source codes for reproducibility.

**Reviewer Confidence:**

2: The reviewer is willing to defend the evaluation, but it is likely that the reviewer did not understand parts of the paper

**Scope:**

4: The work is relevant to the Web and to the track, and is of broad interest to the community

---

### Official Review · Reviewer_uvXZ · 2023-11-23

**Novelty:** 5
**Technical Quality:** 6

**Review:**

### Quality

The authors propose a method CRNTM, which aims to leverage the causal relationships among labels in topic modeling. In CRNTM, both supervised information and latent topics are treated as nodes, with the causal relationships represented as directed edges in a DAG modeled with a Structural Causal Model (SCM). The method design and execution seem sound to me.

The method's performance is evaluated on three datasets and is compared to several (supervised) topic modeling methods. The authors use standard evaluation metrics including topic coherence, topic uniqueness, and topic quality. They also showcase the method's ability to uncover causal relationships between supervised information and latent topics. The experiments are well-executed as well (albeit some additional baselines could be included; see below).

### Clarity

The paper is overall well-written and easy to follow. The figures and tables are also well-presented.

### Originality

The proposed method, CRNTM, is novel based on my knowledge. The major novelty is to explicitly model the causal relationships between supervised information and latent topics.

### Significance

This work is under the category of topic modeling and will facilitate the applications related to topic discovery.

### **Pros**

- The proposed method, CRNTM, is new and interesting.
- The method design is sound.
- The authors provide a thorough evaluation of the method's performance, comparing it to several topic modeling methods.

### **Cons**

- I'm not sure if the statement "casual relationships between supervised information and latent topics have not been visited in prior work" is completely accurate. Although past work may not be under exactly the same setup as studied in this paper, the line of hierarchical topic models can also model topics as DAGs to capture their casual relationships. I would imagine it wouldn't be too hard to extend those methods to incorporate the relationship introduced by supervised information. The paper would benefit from including more discussions of hierarchical topic models.
- I believe CatE (Meng et al.) is also a relevant baseline that can be compared to. I'd encourage the authors to include it in the evaluation.

Reference:
Meng et al. “Discriminative Topic Mining via Category-Name Guided Text Embedding.” WWW 2020.

**Questions:**

Please address the cons raised in my main review.

**Reviewer Confidence:**

3: The reviewer is confident but not certain that the evaluation is correct

**Scope:**

3: The work is somewhat relevant to the Web and to the track, and is of narrow interest to a sub-community

---

### Official Review · Reviewer_i12s · 2023-11-24

**Novelty:** 6
**Technical Quality:** 5

**Review:**

The paper introduces CRNTM, a Causal Relationship-Aware Neural Topic Model, designed to uncover causal relationships within supervised information and latent topics while discovering high-quality topics in neural topic modeling. CRNTM employs a Directed Acyclic Graph (DAG) to represent these relationships, utilizing Structural Causal Models (SCM) to imbue representations with causality. Experimental results demonstrate the reliability and interpretability of the discovered causal relationships, along with the high quality of the learned topics. This novel approach enhances the overall interpretability of neural topic modeling by simultaneously uncovering meaningful causal links and generating coherent and valuable topics.

**Strengths**
- The paper conducts comprehensive experiments across multiple datasets and compares CRNTM with state-of-the-art models, showcasing its superiority in terms of topic quality and interpretability.
- CRNTM not only generates high-quality topics but also provides interpretable causal relationships between supervised information and topics, adding depth to the understanding of topic modeling.

**Areas of Improvement**
- While the paper presents a complex model, some technical components (e.g., SCM) could be explained more intuitively for readers less familiar with causal modeling.

**Questions:**

- I am curious to know what are the specific limitations or edge cases where CRNTM might struggle to establish accurate causal relationships between supervised information and latent topics?

- How sensitive is CRNTM to hyperparameters or variations in the experimental setup, and does it demonstrate robust performance across different types of corpora or domains?

**Reviewer Confidence:**

2: The reviewer is willing to defend the evaluation, but it is likely that the reviewer did not understand parts of the paper

**Scope:**

3: The work is somewhat relevant to the Web and to the track, and is of narrow interest to a sub-community

---

### Official Review · Reviewer_DNFr · 2023-11-29

**Novelty:** 3
**Technical Quality:** 3

**Review:**

Supervised topic modelling has remained a dominant way to model latent topics because the topic models have additional side information in the form of labels that could help guide the latent topic model to generate interpretable topics. Efforts such as those Supervised topic models and maximum margin topic models have been developed a long time ago.

In maximum-margin topic models (https://static.googleusercontent.com/media/research.google.com/en//pubs/archive/38352.pdf), generative and discriminative learning paradigms are exploited. The underlying model is a posterior regularisation framework that regularises the latent space to discover interpretable topics. This paper would have benefitted had such key works been cited.

In this paper, the authors focus not on discovering topics like supervised topic models but on developing causal relationships. The key idea of the model is to jointly discover the causal relationships within a supervised learning framework and discover latent topic information.

From the experimental study, the model improves upon existing methods.

While there are advantages, there are a few disadvantages to this work too.

One advantage is that this work models causal relationships under a supervised learning setting. The key limitation of this work is that it fixes the number of topics rather than finding the number of topics using techniques such as tuning or cross-validation. There is another line of work that not only model word order but also model topic correlations such as the correlated topic model and NTSeg (https://dl.acm.org/doi/pdf/10.1145/2484028.2484062). It would be interesting to find out how this work is different from those works. This is where the discussion on causation and correlation might be important. It is also important that the authors conduct experiments on some downstream applications such as document classification or information retrieval to find out whether the model generalises reliably on downstream applications too. Such experimental analyses have become very popular in the last decade.

**Questions:**

The authors can find the questions in my main comments. I wanted to post questions as I go along my comments to remain coherent.

**Ethics Review Description:**

NIL

**Reviewer Confidence:**

4: The reviewer is certain that the evaluation is correct and very familiar with the relevant literature

**Scope:**

4: The work is relevant to the Web and to the track, and is of broad interest to the community

---

### Decision · Program_Chairs · 2024-01-22

**Decision:**

Accept

**Comment:**

This paper proposes a new neural topic model that is capable of identifying causal relationships among discovered topics. Three components, including pre-encoding, joint encoding, and causal relationship learning, are designed to learn latent topics and corresponding casual relationships. The proposed method is based on sound rationale, and the evaluations are convincing.